



# Development of a Novel Storm Surge Inundation Model Framework for Efficient Prediction

Xuanxuan Gao [1,3], Shuiqing Li [1,2,3,4], Dongxue Mo [1,2,3,4], Yahao Liu [1,2,3,4], and Po Hu [1,2,3,4]

[1]Key Laboratory of Ocean Circulation and Waves, Institute of Oceanology, Chinese Academy of Sciences, Qingdao, 266071, China
[2]Laboratory for Ocean and Climate Dynamics, Pilot National Laboratory for Marine Science and Technology (Qingdao), Qingdao, 266237, China
[3]University of Chinese Academy of Sciences, Beijing, 100049, China
[4]Center for Ocean Mega-Science, Chinese Academy of Sciences, Qingdao, 266071, China

*Correspondence to*: Shuiqing Li (lishuiqing@qdio.ac.cn), Po Hu (hupo@qdio.ac.cn)

**Abstract.** Storm surge is a natural process that generates flood disasters in coastal zone and causes massive casualties and property losses. Therefore, the storm surge inundation is of major concern in formulating appropriate strategies for disaster prevention and mitigation. However, traditional storm surge hydrodynamic models have large limits on computational efficiency and stability in practical applications. In this study, a novel storm surge inundation model was developed based on a wetting and drying algorithm established from simplified shallow water momentum equation. The wetting and drying algorithm was applied to rectangle grid that iterates through cellular automata algorithm to improve computational efficiency. The model, referred to as the Hydrodynamical Cellular Automata Flood Model (HCA-FM), was evaluated by comparing the simulations to regional field observations and that from a widely used hydrodynamic numerical model, respectively. The comparisons demonstrated that HCA-FM can reproduce the observed inundation distributions, and predict consistent results with the numerical simulation in terms of the inundation extent and submerged depth, with much improved computational efficiency (predict inundation within a few minutes) and high stability. The results reflect significant advancement of HCA-FM toward efficient predictions of storm surge inundation and applications at large space scales.

## 1 Introduction

Storm surge, defined as the abnormal water rise (or fall) due to strong winds and/or air pressure gradients, usually tropical cyclones. This coastal water rise combined with the high tide and waves would cause massive inundation. According to the Bulletin of China Marine Disaster, storm surge is the major marine hazard in China, which can cause huge casualties and property damage. Considering the trend of sea level rise and land subsidence, the risk of storm surge inundation in coastal zones is increasingly prominent. Therefore, considerable attention has been given to building accurate and efficient models for the simulation of storm surge inundation, which involves the extent of inundation, submerged depth, and flow velocity whenever technically possible.



Hydrodynamic models that simulate storm surge by numerically solving shallow water equations on a grid (Teng et al., 2017) have been developed and widely used, such as the MIKE21 (e.g., Machineni et al., 2019), the Finite Volume Community Ocean Model (FVCOM) (e.g., Nakamura et al., 2019) and the Advanced Circulation (ADCIRC) model (e.g., Li et al., 2022). These models can also be coupled with nearshore wave models (e.g., Wang et al., 2021).

Aiming to model the storm surge inundation, wetting and drying algorithms have been applied to manage the state transition of elements between wet and dry. The existing algorithms differ in complexity which is related to the computational efficiency, stability and simulation accuracy (Medeiros & Hagen, 2013). The common thread of different algorithms is to design the judgment conditions for grid wetting and drying, which usually uses a criterion by setting a critical value on a physical quantity. The water depth is generally chosen as the physical quantity, with a minimum depth in the wetting and

drying judgment (e.g., Luettich and Westerink, 1999). These numerical inundation model, such as ADCIRC (e.g., Wang et al., 2020; e.g. Shi et al., 2022) and FVCOM (e.g. Saswati Deb et al., 2021), has been used in seveval researches to sucessfully simulate storm surge inundation. These storm surge inundation models embed physics well and thus has high simulation accuracy. However, numerical models have high computational cost and instability due to their numerical complexity and thus are not suitable for applications over large study areas.

Although some efforts have been made to enhance the efficiency (1 to 2 orders of magnitude) of hydrodynamic models through several approaches, such as, neglecting terms in shallow water equations to reduce complexity (e.g., Hunter et al., 2007; Bates et al., 2005; Bates et al., 2010), applying computing technique (e.g., Sanders et al., 2010; Kalyanapu et al., 2011; Vacondio et al., 2017; Roberts et al., 2021), applying subgrid theory (e.g., Volp et al., 2013; Sehili et al., 2014; Kennedy et al., 2019; Begmohammadi et al., 2021; Woodruff et al., 2021), it is still not efficient enough to meet the needs of emergency

applications. Thus, an accurate and efficient storm surge inundation model is urgently needed.

Distinguished from storm surge hydrodynamic models solving shallow water equations, several models for urban flood have been developed for rapid simulation, which could be collectively described as conceptual models. Conceptual models that are mostly used, such as digital elevation model (DEM) based (e.g., Jamali et al., 2018; Manfreda and Samela, 2019; Miura et al., 2021) and Cellular Automata (CA) based models (e.g., Jamali et al., 2019; Wijaya and Yang, 2020, 2021), are

essentially designed for free surface floods. DEM-based models calculate the water distribution above the geographical environment based on the so-called theory the "bathtub method", which assumes a planar water surface. CA-based models have been developed in recent years, and the water volume balance is the most common used basic theory to design the wetting and drying transition rules of CA. Compared with hydrodynamic models, conceptual models have a significant advantage in terms of high computational efficiency, which is crucial to practical applications requiring timeliness. However,

the water distribution in simplified conceptual models is mainly controlled by the force of gravity, which ignore dynamic forces that influence the flood process, such as wind stress and bottom friction, that is important for storm surge inundation. But the basic grid iteration method is worth learning and helps to achieve the design of efficient storm surge inundation model.


In this paper, we propose a novel storm surge inundation model that embeds fluid physics in wetting and drying algorithm and uses CA algorithm to improve computational efficiency. In Sect. 2, the development of the model framework is introduced. Sect. 3 presents model verification and validation against field observations and hydrodynamic model simulations for several typical storm surge inundation events to validate its accuracy by performing comparisons. Finally, in Sect. 4, we analyse the advantages of the model to illuminate its application perspective.

## 2 Model design

### 2.1 Wetting and drying algorithm

The wetting and drying algorithm is the basis of a flooding model (Medeiros & Hagen, 2013), which models spread of water on the computational grid by defining transition rules that govern the state transition of an element between wet and dry. In contrast to transition rules based on the water volume balance in the urban flood model, the transition rules in our model were dynamically designed based on the shallow water momentum equation.

The momentum equation of two-dimensional shallow water equations is given as:

$$\underset{(i)}{\frac{\partial v}{\partial t}} + \underset{(ii)}{v\frac{\partial v}{\partial x}} = \underset{(iii)}{-g\frac{\partial \eta}{\partial x}} + \underset{(iv)}{\frac{\tau_a - \tau_b}{\rho d}} \tag{1}$$

where $v$ is the depth-averaged velocity $(m/s)$ in the direction of $x$, $\eta$ is the water stage $(m)$, $\tau_a$ is the wind stress $(N/m^2)$ in the direction of $x$, $\tau_b$ is the bottom friction $(N/m^2)$ in the direction of $x$, $\rho$ is the seawater density $(kg/m^3)$, $g$ is gravity $(m/s^2)$ and $d$ is the water depth $(m)$.

In Eq. (1), (i) represents the local inertia term, (ii) represents the advective inertia term, (iii) represents the pressure differential term and (iv) represents the external force terms including wind stress and bottom friction.

Neglecting the local inertia term, Eq. (1) is transformed into Eq. (2), which shows the effect of external forces on changing the total energy level.

$$\frac{\partial}{\partial x}\left(\eta + \frac{v^2}{2g}\right) = \frac{\tau_a - \tau_b}{\rho g d} \tag{2}$$

Consider two adjacent cells ($Cell_{tar}$ and $Cell_{ner}$) where $Cell_{ner}$ is wet and $Cell_{tar}$ is dry (Fig. 1). Assume that water will flow from $Cell_{ner}$ to $Cell_{tar}$ which will turn wet. Then, under this null hypothesis Eq. (2) is discretized into Eq. (3) by applying finite differences to act as the linkage between these two cells. We assume that the Froude numbers in the two cells are approximately equal, as expressed in Eq. (4).



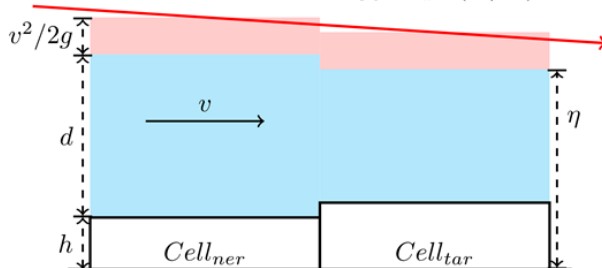

**Figure 1: Illustration of hydrodynamic structure between the target cell and neighbor cell.**

$$\frac{\left[\left(\frac{v_{tar}^2}{2g}+\eta_{tar}\right)-\left(\frac{v_{ner}^2}{2g}+\eta_{ner}\right)\right]}{\Delta x}=\frac{\tau_a-\tau_b}{\rho g d} \tag{3}$$

$$Fr=\frac{v_{tar}}{\sqrt{g(\eta_{tar}-h_{tar})}}=\frac{v_{ner}}{\sqrt{g(\eta_{ner}-h_{ner})}} \tag{4}$$

Thus, Eq. (5) is obtained by combining Eq. (3) and Eq. (4).

$$\eta_{tar}+\frac{Fr^2}{2}(\eta_{tar}-h_{tar})-\left(\eta_{ner}+\frac{v_{ner}^2}{2g}+\frac{\tau_a-\tau_b}{\rho g d}\Delta x\right)=0 \tag{5}$$

However, only if water level $\eta_{tar}$ is higher than the ground elevation $h_{tar}$, is the null hypothesis that water flows from $Cell_{ner}$ to $Cell_{tar}$ made at the beginning true, i.e., that $\eta_{tar}$ is greater than $h_{tar}$, becomes the condition for $Cell_{tar}$ to become wet. Furthermore, combined with Eq. (5), this condition can be transformed into Eq. (6), which describes the relationship between the elevation at $Cell_{tar}$ and the residual height of the energy line after wind forcing and bottom dissipation. Only when the elevation at a dry cell is less than the residual height of the energy line from its adjacent wet cell, will the dry cell be submerged.

$$h_{tar}<\eta_{ner}+\frac{v_{ner}^2}{2g}+\frac{\tau_a-\tau_b}{\rho g d}\Delta x \tag{6}$$

Thus, the water level in $Cell_{tar}$ is obtained by solving Eq. (5) and is expressed as Eq. (7). Then the flow velocity is also calculated as Eq. (8).

$$\eta_{tar}=\left(\eta_{ner}+\frac{v_{ner}^2}{2g}+\frac{\tau_a-\tau_b}{\rho g d}\Delta x+\frac{Fr^2}{2}h_{tar}\right)/\left(1+\frac{Fr^2}{2}\right) \tag{7}$$

$$v_{tar}=Fr\sqrt{g(\eta_{tar}-h_{tar})} \tag{8}$$

The wind stress and bottom friction are given as Eq. (9) and Eq. (11).

$$\tau_a=\rho_a C_d v_{wind}|v_{wind}| \tag{9}$$

$$C_d=\min[(0.75+0.067|v_{wind}|)\times 10^{-3},0.0035] \tag{10}$$

$$\tau_b=\rho C_f v|v| \tag{11}$$

$$C_f=\frac{gn^2}{d^{1/3}} \tag{12}$$

where $\tau_b$ is bottom friction, $\tau_a$ is wind stress, $\rho$ is seawater density, $\rho_a$ is air density, $v_{wind}$ is the projection of relative wind in the direction of water flow, $C_d$ is the drag coefficient suggested by Garratt (1977) with a high limit of 0.0035, $C_f$ is the



bottom friction coefficient, and $n$ is Manning's roughness coefficient, a parameter used to characterize the bed roughness that differs by land cover and can be determined according to the standard of American National Land Cover Data (NLCD)

(Liu et al., 2019).

As bottom friction is reversed to flow, it will always reduce the energy level. The effect of wind force is determined by the relative direction between wind and current. Downstream wind will increase energy level while upstream wind has a similar effect as bottom friction.

## 2.2 Grid model

The wetting and drying algorithm describes the process of water flow from the wet cell to the adjacent dry cell, based on which the storm surge inundation model was built by applying Cellular Automata (CA) algorithm, a grid iteration method widely used in urban flood models.

CA is a mathematical idealized model that can simulate physical systems and processes (Wolfram, 1984). It is discrete spatiotemporally and consists of regular and rigid cells, each of which possesses a set of variables. As the values of variables

at each cell are affected by its neighborhood and updated based on transition rules in discrete time steps, the CA evolves. The flexibility of designing transition rules makes it possible to easily balance the simulation effect and computational efficiency.

In a flood map, the study area can be considered as a binary image, as the state at each position can only be wet or dry. Unlike rainfall-induced inundation, which has no boundary source, storm surge-induced inundation is caused by the

abnormal rise of water along the coastline. As water flows according to the continuity of the fluid, the change in state of any position is only influenced by its neighborhoods within a delay time.




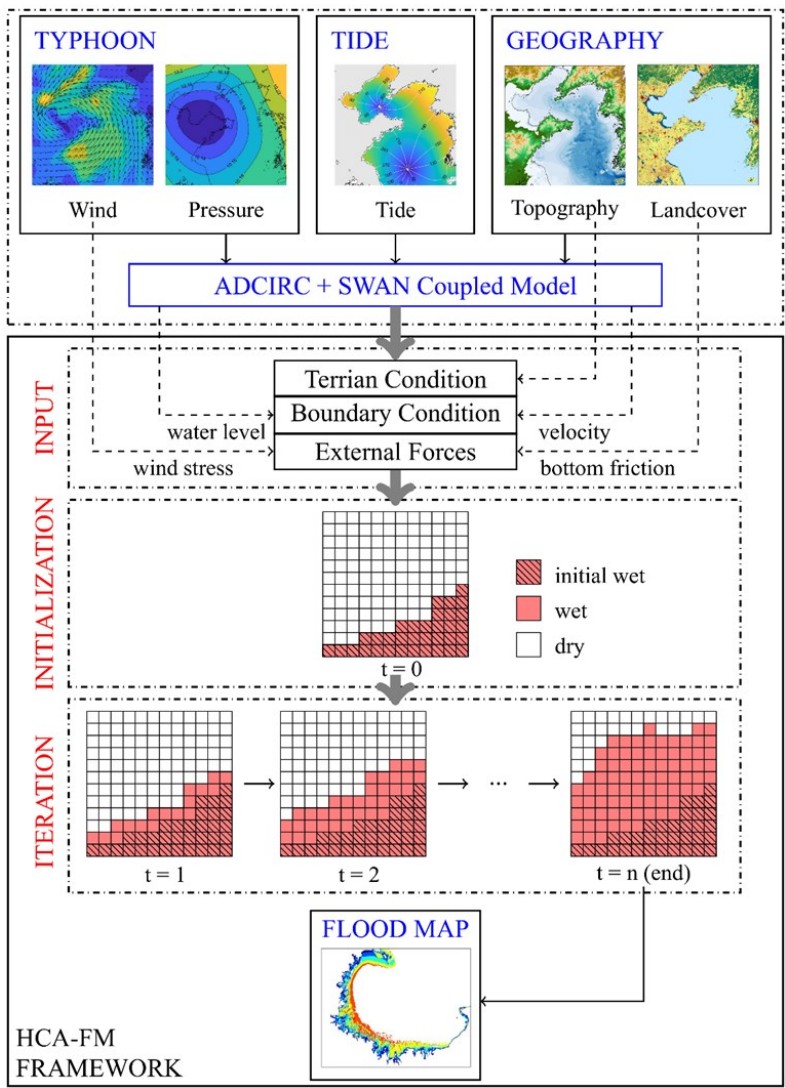

**Figure 2: Framework of storm surge inundation simulation by HCA-FM.**

Taking the wetting and drying algorithm as the transition rule of CA, a novel storm surge inundation model was built. A

simulation by the HCA-FM includes the following steps and an instruction flowchart is also presented in Fig. 2:

(1)  Input data preparation

In the HCA-FM, topography and land cover are used as constant geographic features in the study area. Land cover type will be transformed to Manning's coefficient according to NLCD to determine the bottom friction. The bottom friction together with wind stress are external forces which determine the energy slope between cells. Boundary conditions that

include water level and flow velocity along coastline are used as triggers for storm surge inundation.

(2)  Initialization



The study area is divided into rectangle cells with suitable resolution to maintain a good balance between spatial resolution and computational efficiency. The neighborhood is defined as the Moore neighborhood, which includes the cell itself, four cells with a common edge and four cells with a common vertex.

The ground elevation, Manning coefficient, and wind field are resampled to all cells while coastal water level and flow velocity are resampled to the initial wet cells. Cells with water levels larger than elevation are given the state of wetness expressed by 255 in the program while others are dry expressed by 0. Through these preparatory works, the initial cell structure of the HCA-FM is built.

(3) Grid iteration

Then each cell in the grid will iterate according to the wetting and drying algorithm introduced in Sect. 2.1. Briefly, in at least one of the eight directions, if the relationship between the target cell and wet neighbor cell satisfies Eq. (6), the target cell will turn wet. If more than one wet neighbor cell meets this criterion, the HCA-FM will regard them as one cell by averaging the residual energy height and Froude number. Then the submerged depth and flow velocity will be calculated by Eq. (7) and Eq. (8) respectively. These cells will be updated in the next iteration step, and others will remain. The same process will be performed for the whole grid step by step until no cell should be updated.

Following these three steps, the HCA-FM achieves simulation of storm surge inundation. The final inundation extent, level of submerged depth and flow velocity are obtained from the final set of wet cells, which describe the extent and degree of hazard.

## 3 Model assessment

### 3.1 Storms and area

The assessment of the HCA-FM was based on two aspects: validation against field observations and comparison with a numerical hydrodynamic model. Two typhoon storm surge cases were included in the experiments for each aspect.

In the validation against field observations, coastal zones of Cangzhou, Hebei and Shenzhen, Guangdong were selected as study regions and represent typhoon-induced storm surges that occurred in the Bohai Sea and the South China Sea, respectively. Typhoon Lekima (Fig. 3(a)) landed on Shandong Peninsula on October 11[th], 2019, with a center maximum wind speed of grade 9 and increased severe storm surge in Bohai Bay and Laizhou Bay. The National Marine Environmental Forecasting Center organized teams to investigate disasters around the south coast of Bohai Bay. The approximate inundation extent in Cangzhou was obtained. Typhoon Hato (Fig. 3(b)) originated in the northwestern Pacific Ocean on August 20[th], 2017. Due to its movement track and intensity, the Pearl River Delta region experienced strong winds and severe storm surges. The Marine Monitoring and Forecasting Center of Shenzhen organized teams to investigate disasters in key regions. The field survey data in Shenzhen after Typhoon Hato included several locations that underwent hard-hits, which could indirectly reflect the inundation extent. Due to the limitations of rough survey data, validation only relied on inundation extent.





In the validation against the numerical model, Laizhou Bay was selected as the study area, which is located south of Bohai

and frequently experiences storm surges. Due to its unique geographic site and configuration, the magnitude of the surge level along the coastline of Laizhou Bay is higher than that along any other part of the Shandong Peninsula. As a result, coastal regions in Laizhou Bay are exposed to a more severe situation where enormous casualties and economic losses are more likely to occur. Lekima and Polly were two typhoon storm surge events that caused severe consequences around Laizhou Bay (Fig. 3(a)). Along the coastline of Laizhou Bay, Typhoon Lekima brought about a surge rise of approximately

150 to 200 cm, which loaded heavy risk on the coastal regions. Different from Lekima, of which the typhoon center passed directly through Laizhou Bay, Typhoon Polly passed through the south of Shandong Peninsula. Therefore, these two typhoon processes could represent different types of typhoon tracks to check the universality of the HCA-FM.

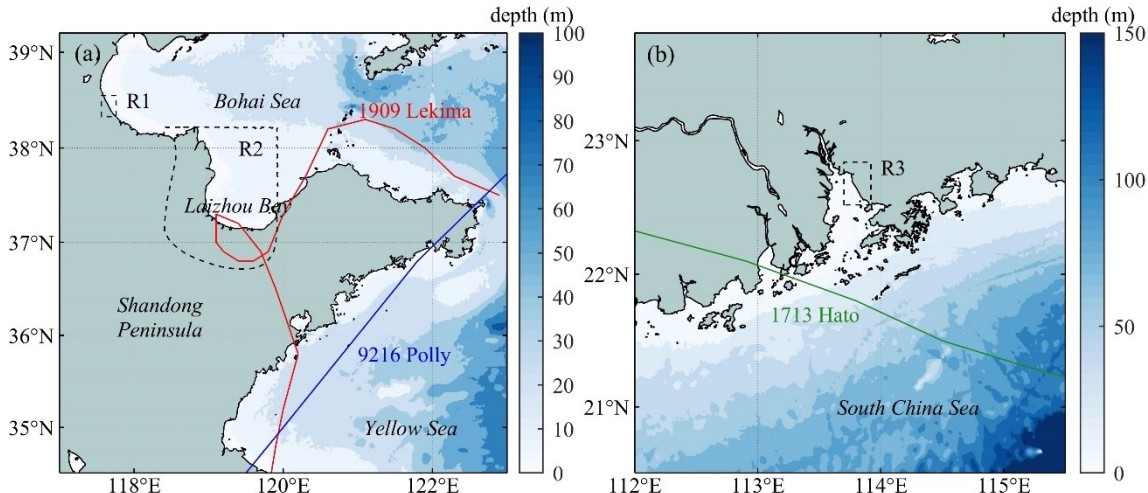

**Figure 3: Typhoon tracks, bathymetry and study regions. (a) track of Lekima (1909) and Polly (9216); R1: Cangzhou, Hebei; R2:**
**Laizhou Bay; (b) track of Hato (1713); R3: west coast of Shenzhen, Guangdong.**

The HCA-FM was performed at Cangzhou during Lekima and the west coast of Shenzhen during Hato to validate the model simulations with field observations. Then, the HCA-FM and a hydrodynamic model called ADvanced CIRCulation + Simulating WAves Nearshore (ADCIRC+SWAN) coupled model were performed in the coastal region of Laizhou Bay during Lekima and Polly to make comparisons and validate the simulation accuracy of the HCA-FM. For the inundation

simulation, the inundation extent and submerged depth are the aspects of greatest concern that determine the degree of ultimate consequence. Thus, these two aspects were chosen as indicators to measure the simulation consistency between the two models. ADCIRC+SWAN also provided boundary inputs, including the surge level and flow velocity, for the HCA-FM. Datasets that support ADCIRC+SWAN and HCA-FM model are available online and their sources are shown in Table 1.

**Table 1: Sources of datasets that support ADCIRC+SWAN and HCA-FM model.**

| Name | Data sources |
| --- | --- |
| Bathymetric data | General Bathymetric Chart of the Oceans (GEBCO; https://www.gebco.net/) |



| Topographic data | ASTER GDEM v2 (The data set is provided by Geospatial Data Cloud site, Computer Network Information Center, Chinese Academy of Sciences; http://www.gscloud.cn) |
|---|---|
| Wind and pressure field | CFSR (from 1979 to 2010, Saha et al., 2010; https://rda.ucar.edu/datasets/ds093.1/) data from NECP<br>CFSv2 (from 2010 to now, Saha et al., 2011; https://rda.ucar.edu/datasets/ds094.1/) data from NECP |
| Land cover | GlobeLand30 from National Geomatics Center of China (Chen et al, 2014; https://www.ngcc.cn/) |
| Typhoon tracks | China meteorological administration (Ying et al., 2014; Lu et al., 2021; http://tcdata.typhoon.org.cn) |

## 3.2 Comparison with field observations

Based on the field observations in Cangzhou after Typhoon Lekima and Shenzhen after Typhoon Hato, the HCA-FM was performed for these two cases. The boundary surge level and flow velocity were set as the temporal maximum conditions during storm surge processes. The inundation extents simulated by the HCA-FM and survey extents are presented in Fig. 4.

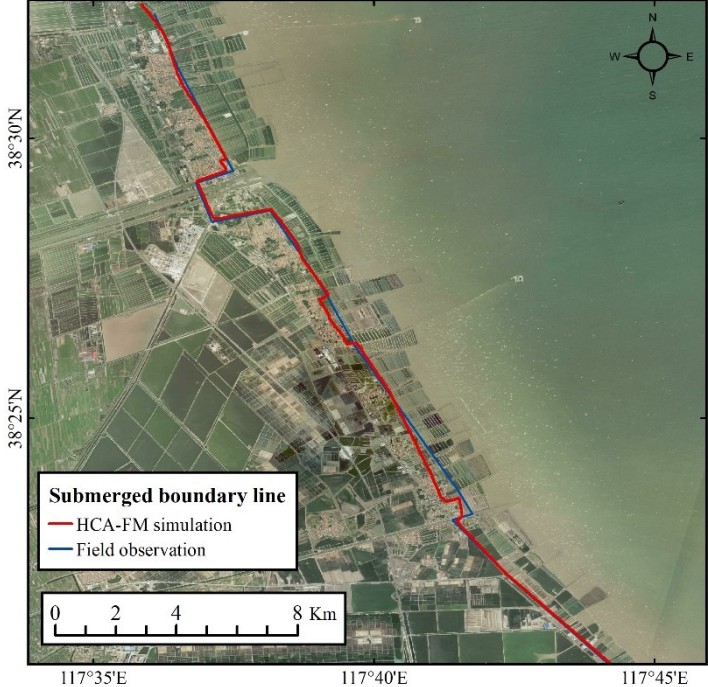

**Figure 4: Comparison of inundation between the HCA-FM simulation and field observation in Cangzhou, Hebei during Typhoon Lekima; red line represents submerged boundary line simulated by the HCA-FM and blue line by field survey (Base map by Esri, Maxar, Earthstar Geographics, and the GIS User Community).**





As shown in Fig. 4, the submerged boundary line simulated by the HCA-FM is basically consistent with field observations.
The coastal region of Cangzhou is characterized by gentle terrain, waterways, and human-made seawalls. Most segments of

the submerged boundary line are determined by seawalls because of the low-laying terrain. The flection of the submerged
boundary line landwards is located near rivers, through which storm surge water rise spreads toward land more easily and
causes further inundation compared with other regions. In other words, the HCA-FM can reflect the natural topography and
human-made projects which enables simulation under a real geographical environment.

Figure 5 displays the comparison of the inundation extent between the simulation and field observations on the west coast of

Shenzhen during Typhoon Hato. Overall, Hato brought heavy disasters to the west coast of Shenzhen. Seawater intruded
landward for several kilometers due mainly to low-lying elevation and water intrusion backward through waterways. The
detailed comparisons of three survey positions are presented as three subplots of Fig. 5.

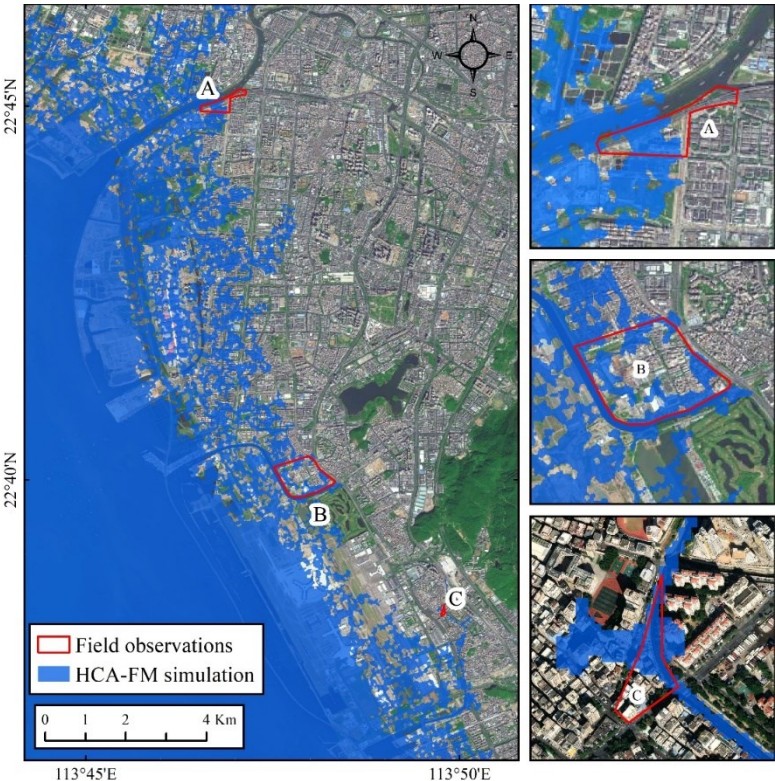

**Figure 5: Comparison of inundation in three survey sites (A, B, C) on the west coast of Shenzhen, Guangdong during Typhoon**
**Hato; blue region represents flood extent simulated by the HCA-FM and red polygons represent survey regions being submerged**
**(Base map by Esri, Maxar, Earthstar Geographics, and the GIS User Community).**

Considering the errors between the digital terrain model (DEM) and real terrain, the differences between HCA-FM
simulation and survey data are acceptable. In additional, rainfall, which is not considered in the current HCA-FM, is another
influencing factor that is blamed for generating the differences. The accuracy of the HCA-FM was validated. Moreover,





these two comparison cases located on the coast of the Bohai Sea and the South China Sea also indicated the universality of the HCA-FM.

### 3.3 Comparison with Hydrodynamic Model

The numerical hydrodynamic model can represent the basic fluid mechanics well and thus is used here to assess the physical consideration of the HCA-FM. In reality, seawalls are built in coastal regions to prevent surge inundation. In this section, to focus on the inundation process, seawalls were neglected. This comparison experiment focused on the region around Laizhou Bay, during two storm surge events forced by Typhoons Lekima and Polly, respectively.

The numerical hydrodynamic model was chosen as the widely used the ADCIRC+SWAN coupled model. The ADCIRC model was developed by Luettich et al. (1992) and Westerink et al. (1994), and SWAN model was the third-generation wave model developed by Delft University of Technology (Booij et al., 1999). The ADCIRC+SWAN model couples the ADCIRC and SWAN models by swapping data in the same grid and is generally applied for the simulation of tropical cyclone-induced storm surge and coastal inundation (Dietrich et al., 2011). The grid resolution of ADCIRC+SWAN model used in the comparison experiments ranges from 100 m on the inland coast to 20 km along the abyssal open boundary. The surge simulation ability of ADCIRC+SWAN model was well verified by tide station data (Li et al., 2023).

The input data of the maximum surge level and flow velocity at the initial wet cells in the HCA-FM were extracted from the ADCIRC+SWAN model to unify the boundary conditions. In addition to boundary conditions, bottom friction parameterizations can also influence model's performance (Chen et al., 2013). Considering that the bottom friction coefficient is defined as Eq. (13) in the ADCIRC+SWAN model, while defined as Eq. (12) in HCA-FM in the form of Manning's coefficient, it is necessary to use an equivalent bottom friction model in the experiments.

$$C_f = C_{f_{min}} \left[ 1 + \left( \frac{H_{break}}{d} \right)^{\theta} \right]^{\frac{\gamma}{\theta}} \tag{13}$$

where $C_f$ is the bottom friction coefficient, $C_{f_{min}}$ is the minimum bottom friction coefficient, $H_{break}$ is the break depth, $d$ is the water depth, $\theta$ is a dimensionless parameter that defines how rapidly the bottom friction coefficient approaches its upper and lower limits, and $\gamma$ is a dimensionless parameter that describes how quickly the bottom friction coefficient increases as the water depth decreases. $C_{f_{min}}$, $H_{break}$, $\theta$, and $\gamma$ are set as 0.0015, 1, 10 and 1/3, respectively, in the ADCIRC+SWAN model.

Based on the relationship between $C_f$ and $n$ in Eq. (12), an equivalent Manning's coefficient, as expressed in Eq. (14), was used in HCA-FM, which is related to water depth instead of land cover type.

$$n = \sqrt{\frac{0.0015 d^{\frac{1}{3}}}{g} \left[ 1 + \left( \frac{1}{d} \right)^{10} \right]^{\frac{1}{30}}} \tag{14}$$

Then, the extent and submerged depth of inundation during Typhoons Lekima and Polly around Laizhou Bay were simulated by the HCA-FM and ADCIRC+SWAN models respectively. Considering that different grid structure were used in these two





models (orthogonal grids in HCA-FM while unstructured triangular grids in ADCIRC+SWAN model) and the grid resolution of HCA-FM was higher than the ADCIRC+SWAN model, the water depth simulated by HCA-FM was interpolated to the grid nodes of ADCIRC+SWAN model by bilinear interpolation to display water depth comparisons.

As a result, the comparison analysis of inundation results by the two models consisted of the following two aspects:

(1) A comparison of inundation extent was performed to demonstrate the accuracy of the simulated inundation range. The
statistic used to assess the degree of consistency between the two models' simulated inundation extent was defined as the fit ratio $\delta$ (Horritt and Bates, 2001). $\delta$ ranges from 0 for no overlap to 1 for perfect fit.

$$\delta = \frac{N_o}{N_a + N_m - N_o} \tag{15}$$

where $N_a$ is the number of wet points in the HCA-FM, $N_m$ is the number of wet points in the ADCIRC+SWAN model, and $N_o$ is the number of overlapping wet points.

(2) A comparison of the submerged depth at the overlapping wet points was performed to demonstrate the accuracy of the simulated submerged depth. Statistics to assess the consistency of the two groups of submerged depths simulated by the two models included R square ($R^2$) and root-mean-square errors (RMSE).

$$R^2 = \left[ \frac{\sum_{i=1}^{N_o}(d_a^i - \overline{d_a})(d_m^i - \overline{d_m})}{\sqrt{\sum_{i=1}^{N_o}(d_a^i - \overline{d_a})^2 \cdot \sum_{i=1}^{N_o}(d_m^i - \overline{d_m})^2}} \right]^2 \tag{16}$$

$$RMSE = \sqrt{\frac{1}{N_o}\sum_{i=1}^{N_o}[d_a^i - d_m^i]^2} \tag{17}$$

where $d_a^i$ is the water depth at point $i$ simulated by the HCA-FM, and $d_m^i$ is the water depth at point $i$ simulated by the ADCIRC+SWAN model.

Figure 6 shows the comparisons of simulated inundation between the HCA-FM and ADCIRC+SWAN models during Typhoons Lekima and Polly. Figure 6(a) and 6(c) displays the maximum inundation extent during Lekima and Polly by the two models. The green region is regarded to be submerged by both models, while the red and blue regions represent the
overestimated and underestimated inundation areas, respectively, simulated by the HCA-FM compared with the ADCIRC+SWAN model. It was intuitively plausible that the HCA-FM simulated a basically consistent inundation extent with the ADCIRC+SWAN model. The fit ratio was 0.92 for Lekima and 0.95 for Polly, which was in accordance with intuition. Figure 6(b) and 6(d) displays the comparison of the submerged depth point-to-point simulated by the two models within the overlapping area during Lekima and Polly. Most of the dots are distributed around $y = x$ within an error of 0.3 m,
the values of $R^2$ was 0.96 and $RMSE$ was 0.13 m for Lekima ($R^2$ was 0.96 and $RMSE$ was 0.12 m for Polly), indicated a good consistency between the HCA-FM and ADCIRC+SWAN models.



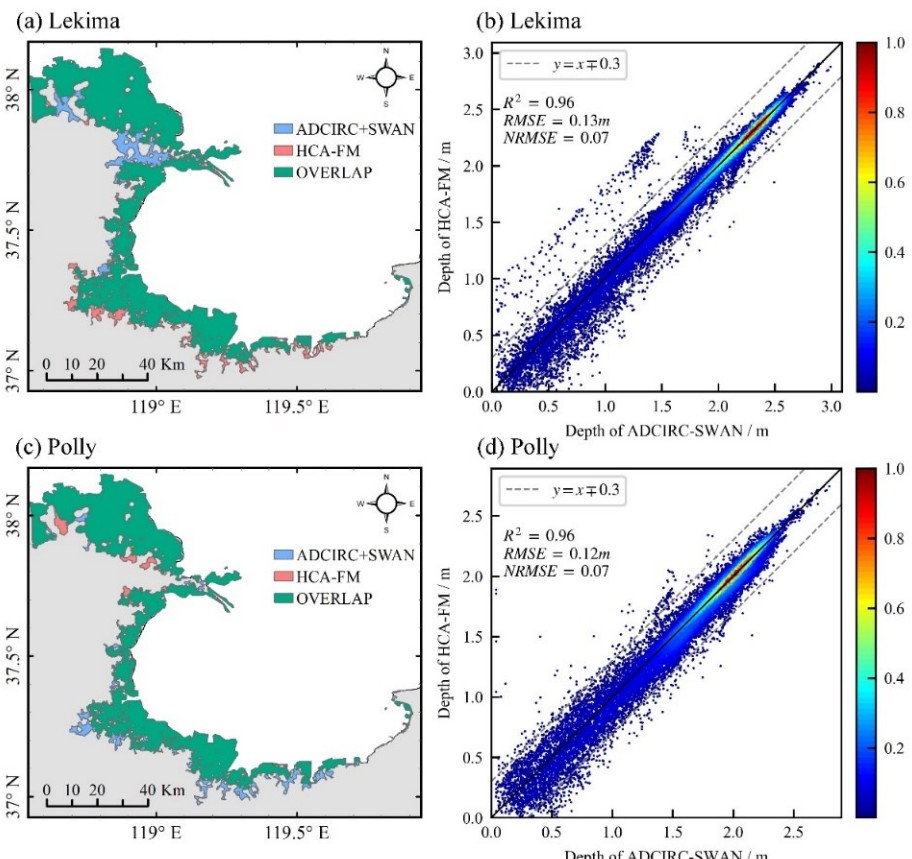

**Figure 6: Comparison between the HCA-FM and ADCIRC+SWAN models. (a) & (c) Comparison of flood extent during Typhoons Lekima and Polly, respectively; (b) & (d) scatter plot of ADCIRC+SWAN versus HCA-FM simulated water depth during Lekima and Polly, respectively.**

## 4 Discussions

Inundation models aim to serve realistic hazard management. Thus, it is necessary for them to provide flood maps that are sufficiently accurate. Image comparisons on inundation extent polygons were applied, and model simulations can reflect the real inundation for both storm surge events. Such results show the simulation accuracy of the HCA-FM under a real geographical environment, where seawalls and rivers are considered. On the other hand, image and indicator comparisons were applied to validate the model's accuracy and theoretical rationality in contrast to the ADCIRC+SWAN coupled model. With the fit ratio $\delta$ higher than 0.90, $R^2$ higher than 0.95 and $RMSE$ less than 0.15 m, a conclusion could be made that the error of the simulated inundation extent and submerged depth compared with the ADCIRC+SWAN model is basically small and acceptable. This acceptable error is relative to the grid structure difference (Tsubaki and Kawahara, 2013), model's simplified hydrodynamic basis, and simplifications on boundary conditions (Parizi et al., 2022).





According to the assessment results, the accuracy of the HCA-FM is highly beneficial from its hydrodynamic consideration, of both wind force and bottom friction. To illustrate this positive effect on simulation accuracy, sensitivity experiments associated with wind force and bottom friction for two storm surge events were designed.

Table 2 shows the consistency indicators of the four groups of sensitivity experiments. Experiment (Exp.) 1 represents the
standard HCA-FM with consideration of both wind force and bottom friction, which provides the good simulation discussed above. Experiment 2 represents the model without consideration of wind force and bottom friction as many conceptual models do and provides a worse simulation with the $R^2$ less than 0.90 and $RMSE$ higher than 0.25 m. This comparison between Exp. 1 and Exp. 2 demonstrates the necessity of considering forces affecting water bodies. Experiment 3 and Exp. 4 represent experiments in which only wind force or bottom friction is considered, and both provide worse results compared
with Exp. 1, among which Exp. 4, which considers bottom friction but ignores wind force, provides an even worse simulation. This finding shows that consideration of external forces improves the model's accuracy and that bottom friction plays a more dominant role than wind stress in water propagation due to its important work in energy dissipation (Akbar et al., 2017; Chu et al., 2019).

**Table 2: Consistency of simulated inundation around Laizhou Bay between the HCA-FM and ADCIRC+SWAN models in**
**sensitivity experiments against wind force and bottom friction.**

|  |  | Exp. 1 | Exp. 2 | Exp. 3 | Exp. 4 |
|---|---|---|---|---|---|
| Wind force |  | on | off | on | off |
| Bottom friction |  | on | off | off | on |
| Lekima (1909) | $\delta$ | 0.92 | 0.89 | 0.91 | 0.84 |
|  | $R^2$ | 0.96 | 0.89 | 0.93 | 0.75 |
|  | $RMSE$ (m) | 0.13 | 0.27 | 0.26 | 0.44 |
| Polly (9216) | $\delta$ | 0.95 | 0.91 | 0.91 | 0.87 |
|  | $R^2$ | 0.96 | 0.83 | 0.92 | 0.66 |
|  | $RMSE$ (m) | 0.12 | 0.28 | 0.24 | 0.45 |

*Note.* **Here $\delta$, $R^2$, and $RMSE$ represents the difference between HCA-FM and ADCIRC+SWAN simulations, the fit ratio of inundation range, R square and root-mean-square error of inundation depth, respectively.**

These comparisons result in the conclusion that the HCA-FM provides simulation consistent with the ADCIRC+SWAN model and is superior to most existing conceptual urban flood models. In addition to the accuracy of the HCA-FM, this
simulation method's advantage also lies in its computational efficiency due to the use of a CA algorithm compared with a numerical model. As shown in Table 3, for the simulations of storm surge inundation during Typhoon Lekima, the ADCIRC+SWAN coupled model taken 137843 s in CPU time while the HCA-FM model only taken 39.3 s. The reduction in computational time as well as its high stability and universality would have great significance in the practical applications of storm surge inundation simulation.



**Table 3: Grid type, resolution, number of cells and run time (CPU time) for the 1909 Lekima storm surge inundation simulations by HCA-FM and ADCIRC+SWAN model.**

| Model | Grid type | Resolution (m) | cells | Run CPU time (s) |
|---|---|---|---|---|
| ADCIRC+SWAN | triangle | 100–1000 | 91290 | 137843 |
| HCA-FM | rectangle | 88 × 111 | 2301145 | 39.3 |

## 5 Summary

In this paper, a novel storm surge inundation model (HCA-FM) based on a wetting and drying algorithm derived from the simplified shallow water momentum equation was proposed for quick and accurate simulation. The model triggered by

boundary water level and flow velocity is built with an energy perspective considering the dominant impact factors including wind force and bottom friction. Moreover, the use of CA algorithm greatly inproves the computational efficiency and computational stability. Credible comparison results to field observations for different regions and typhoon processes verified the model's accuracy in predicting the maximum flood extent and depth.

The HCA-FM reconciles computation cost and physical considerations when compared to other numerical and conceptual

models. It is superior to traditional conceptual models in reflecting the hydrodynamic characteristics of storm surge inundation, and it also demonstrates its superiority over numerical models by significantly improving the computation efficiency. Therefore, the HCA-FM is a more appropriate candidate for predicting storm surge inundation in practical use.

## Data availability

Experimental datasets in this paper are available at http://doi.org/10.5281/zenodo.10596631 (Gao et al., 2024). The HCA-

FM codes and instruction file are available at http://doi.org/10.5281/zenodo.10596826 (Gao, 2024). Data sets including geographical, hydrological, and meteorological data that support ADCIRC+SWAN and DCA-FM model are publicly available, including the General Bathymetric Chart of the Oceans (GEBCO; https://www.gebco.net/), the ASTER GDEM V2 provided by Geospatial Data Cloud site, Computer Network Information Center, Chinese Academy of Sciences (http://www.gscloud.cn), the NCEP Climate Forecast System Reanalysis (CFSR, from 1979 to 2010, Saha et al., 2010;

https://rda.ucar.edu/datasets/ds093.1/) and Version 2 (CFSv2, from 2010 to now, Saha et al., 2011; https://rda.ucar.edu/datasets/ds094.1/) Seclected Hourly Time-Series Products, and the GlobeLand30 land cover data openly available in National Geomatics Center of China (Chen et al, 2014; https://www.ngcc.cn/). The typhoon tracks are from the China Meteorological Administration tropical cyclone database (Ying et al., 2014, Lu et al., 2021; https://tcdata.typhoon.org.cn/).



## Author contributions

XG, SL, and PH contributed to model's conceptualization, methodology; XG contributed to code development and writing of manuscript draft; SL reviewed and edited the manuscript; DM and YL helped with the methodology.

## Competing interests

The authors declare that they have no conflict of interest.

## Acknowledgments

This research was sponsored by the National Key Research and Development Program of China (2023YFC3008203), the National Natural Science Foundation of China (42076214, 41976010, U1706216, U1806227).

We would like to acknowledge the ADCIRC+SWAN coupled model developed by the ADCIRC development team. We thanked the has data support provided by NCEP CFS, GEBCO, and CMA. We appreciated the technical support given by and the High Performance Computing Center, IOCAS and the data service provided by the Oceanographic Data Center, Chinese Academy of Sciences (CASODC) (http://msdc.qdio.ac.cn).

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
