# Peer review of "Development of a Novel Storm Surge Inundation Model Framework for Efficient Prediction"

_Geoscientific Model Development, 2024_

## Author Response (AR2)

Dear Topic editor,

We greatly appreciate the time and effort that the editorial team and reviewers have put into evaluating our manuscript. These comments are all valuable and very helpful for revising and improving our work, as well as providing important guidance for our research. We have carefully studied the comments and have made revisions that we hope meet with approval. Here we provide a detailed point-by-point response to each reviewer's comments and list all changes in the revised manuscript.

**1. Comments from referees**

**RC1 from Anonymous Referee #1**

(1) Can authors explain the limitations of traditional storm surge hydrodynamic models that prompted the development of the Hydrodynamical Cellular Automata Flood Model (HCA-FM), particularly regarding computational efficiency and stability?

(2) How does the wetting and drying algorithm derived from simplified shallow water momentum equations contribute to the computational efficiency and stability of the HCA-FM model?

(3) Could authors elaborate on how the rectangle grid and cellular automata algorithm are utilized within the HCA-FM model to improve computational efficiency, and what advantages does this approach offer over traditional methods?

(4) In evaluating the HCA-FM model, what were the critical criteria used to assess its performance, and how did the model compare to regional field observations and simulations from a widely used hydrodynamic numerical model?

(5) Can authors provide insights into validating the HCA-FM model against regional field observations, including the specific data sources and methodologies used for comparison?

(6) What were the main findings regarding the ability of the HCA-FM model to reproduce observed inundation distributions, and how did its predictions compare to those of the hydrodynamic numerical model regarding inundation extent and submerged depth?

(7) The study mentions significantly improved computational efficiency with the HCA-FM model, allowing for inundation predictions within a few minutes. Can authors discuss the implications of this improved efficiency for real-time storm surge forecasting and disaster management?

(8) How does the high stability of the HCA-FM model contribute to its reliability in predicting storm surge inundation, particularly under varying environmental conditions and input parameters?

(9) Considering the significant advancements demonstrated by the HCA-FM model, what are the potential applications at large space scales, and how might they contribute to more effective disaster prevention and mitigation strategies in coastal regions?

(10) Lastly, based on the findings of this study, what are the critical areas for future research or refinement of the HCA-FM model, and how might it be further optimized for broader practical use in storm surge prediction and coastal zone management?

**RC2 from Anonymous Referee #1**

The authors have well addressed my comments. This manuscript is acceptable for publication in its present form.

**RC3 from Anonymous Referee #2**

The developed model, called HCA-FM, uses a different approach for the wet-drying algorithm compared to the common CA-flood models, which typically employ the water balance approach. The novelty of this manuscript lies in the fact that HCA-FM utilizes the shallow water momentum equation to govern the transition rules for the CA model and incorporates external forces (e.g., wind stress and bottom friction). Therefore, the main focus of this study is to prove that by adding these new rules or equations, it will improve the accuracy of HCA-FM compared to common CA-flood models and its efficiency compared to conventional hydrodynamic models. However, based on this manuscript, the authors seem to fail to demonstrate this. The authors directly applied the model to a real event and compared the results with the ADCIRC+SWAN model. In my opinion, the authors could have used simpler examples (such as laboratory experiments or artificial cases) to prove the hypothesis. By using simpler cases, we can observe in detail what is happening within the model's results. From what I have seen, since the authors directly applied the model, we can only observe the comparison of flood extent between the model's results and observational data; therefore, the discussion is not deep enough.

2.  **Author's response and changes in manuscript to Comments from Anonymous Referee #1**

**RC1 from Anonymous Referee #1**

**(1) RC1-Q1: Can authors explain the limitations of traditional storm surge hydrodynamic models that prompted the development of the Hydrodynamical Cellular Automata Flood Model (HCA-FM), particularly regarding computational efficiency and stability?**

**Response:** Traditional storm surge hydrodynamic models, such as the ADCIRC model, simulate the storm surge inundation by numerically solving full shallow water equations. However, it is hard to balance scientific accuracy, numerical stability, and computational efficiency in computer modeling. The reasons are detailed as follows: High Computational Cost: The time step of the hydrodynamic model is restricted by the grid resolution to ensure stability, because the stability of the solution is limited by the CFL restriction on the gravity wave speed ($\sqrt{gH}\Delta t/\Delta x < C_r$, in which a practical $C_r$ upper bound of is 0.5 or smaller). To guarantee the spatial accuracy and computational stability of the storm surge inundation simulation, the grid resolution must be increased, and the time step must be reduced to ensure model stability, which significantly increases the computational cost.

As a result, most hydrodynamic modeling applications require significant computational resources. In addition, many complicated preparations must be done before using a hydrodynamic model, particular for a unstructured mesh, including mesh generation, mesh quality adjustment, and input file generation. It is not convenient for practical use. These limitations have prompted the development of the Hydrodynamical Cellular Automata Flood Model (HCA-FM) to provide a computationally efficient and easy-to-use storm surge flood model.

**Revisions in manuscript (RC1-Q1):** These limitations of traditional storm surge hydrodynamic models have been supplemented in the introduction to make the motivation for HCA-FM model developing clearer:

(Page 2, line 43-47) However, it is hard to balance scientific accuracy, numerical stability, and computational efficiency in computer modeling. To guarantee the spatial accuracy and computational stability of the storm surge inundation simulation, the grid resolution must be increased, and the time step must be reduced to ensure model stability, which significantly increases the computational cost. numerical models have high computational cost and instability due to their numerical complexity and thus are not suitable for applications over large study areas.

(Page 2, lines 53-55) And it is not convenient for practical use as many complicated preparations must be done before using a hydrodynamic model, particular for a unstructured mesh, including mesh generation, mesh quality adjustment, and input file generation.

**(2) RC1-Q2: How does the wetting and drying algorithm derived from simplified shallow water momentum equations contribute to the computational efficiency and stability of the HCA-FM model?**

**Response:** The traditional numerical models require a significant amount of time to solve hydrodynamic equations using discretization methods. However, in HCA-FM, we transform the shallow water equation into a straightforward wetting and drying algorithm and dynamic calculation formula and the amount of computation is greatly reduced. Based on this, we have developed the efficient grid iteration algorithm of HCA-FM using cellular automata. Therefore, on the premise of ensuring the accuracy of the calculation results, this approach improves computational efficiency and avoids instability that may arise when solving complex equations using discretization methods.

The relevant content has been supplemented in the manuscript to make the description clearer.

**Revisions in manuscript (RC1-Q2):** We added a description of the main determining factor on the computational efficiency of CA, to illustrate the contribution of wetting and drying algorithm derived from simplified shallow water momentum equations and CA algorithm that improve the computational efficiency of the HCA-FM model:

(Page 16-17, lines 343-349) In addition to the accuracy of the HCA-FM, this simulation method's advantage also lies in its computational efficiency due to the simplified wetting and drying algorithm (transition rules) derived from the shallow water equations.  In comparison, traditional methods like numerical model solve hydrodynamic equations using discretization methods and thus require a significant amount of time. As shown in Table 3, for the simulations of storm surge inundation during Typhoon Lekima, the ADCIRC+SWAN coupled model taken 137843 s in CPU time while the HCA-FM model only taken 39.3 s.

**(3) RC1-Q3: Could authors elaborate on how the rectangle grid and cellular automata algorithm are utilized within the HCA-FM model to improve computational efficiency, and what advantages does this approach offer over traditional methods?**

**Response:** A cellular automata (CA) is typically composed of a group of cells that represent a discretized space, each of which has a state, a distribution of neighboring cells, a discrete time step, and a set of transition rules. The transition rules dominate the update process of CA by determining the new state of each cell in terms of its current state and the states of the cells in its neighborhood. The computational efficiency of the CA algorithm depends on the complexity of the transition rules.

The wetting and drying algorithm (transition rules) of HCA-FM, which is derived from the simplified shallow water equations, is very simple and therefore computationally efficient. In comparison, traditional methods like numerical model solve hydrodynamic equations using discretization methods and thus require a significant amount of time. Therefore, HCA-FM has an advantage in terms of computational efficiency.

Rectangular grid used in HCA-FM belongs to structural grid, which is ordered. Structural grids can be generated quickly with high mesh quality without the need for mesh quality control. In structural grids, the neighborhood of

each node can be obtained automatically according to the grid numbering rules. Therefore, the data structure of structural grids is simple and does not require specific storage, thus effectively saving computational memory. The use of structural grid used in HCA-FM simplifies the preparatory work and shortens the run time.

The relevant content has been supplemented in the manuscript to make the description clearer.

**Revisions in manuscript (RC1-Q3):** We added a description of the main determining factor on the computational efficiency of CA, to illustrate the contribution of wetting and drying algorithm derived from simplified shallow water momentum equations and CA algorithm that improve the computational efficiency of the HCA-FM model:

(Page 5, lines 131) The computational efficiency of the CA algorithm depends on the complexity of the transition rules.

(Page 16-17, lines 343-349) In addition to the accuracy of the HCA-FM, this simulation method's advantage also lies in its computational efficiency due to the simplified wetting and drying algorithm (transition rules) derived from the shallow water equations. use of a CA algorithm compared with a numerical model. In comparison, traditional methods like numerical model solve hydrodynamic equations using discretization methods and thus require a significant amount of time. As shown in Table 3, for the simulations of storm surge inundation during Typhoon Lekima, the ADCIRC+SWAN coupled model taken 137843 s in CPU time while the HCA-FM model only taken 39.3 s. The reduction in computational time as well as its high stability and universality would have great significance in the practical applications of storm surge inundation simulation.

**(4)  RC1-Q4: In evaluating the HCA-FM model, what were the critical criteria used to assess its performance, and how did the model compare to regional field observations and simulations from a widely used hydrodynamic numerical model?**

**Response:** In the comparison of HCA-FM simulations with field observations (Section 3.2), the visual image comparisons were made for the line or polygon features of inundation extent. The experiment contains two typhoon storm surge processes, and the study areas are Cangzhou, Hebei and Shenzhen, Guangdong. The model performance is tested by the fitness of line or polygon features of inundation extent by visual comparison. These have been described in the lastest version of the manuscript (Section 3.2), but supplements and modifications are made in the revised manuscript to make the description clearer.

In the experiment of comparing simulations between HCA-FM and numerical simulations from ADCIRC+SWAN model (Section 3.3), comparisons were made for inundation extent and depth in Laizhou Bay for two typhoon storm surge processes (Lekima and Polly). The two models are using the same topo data. The index of agreement in the comparison includes the fit ratio δ of inundation extent, squared correlation coefficient $R^2$ and root mean squared error RMSE for water depth. The fit ratio δ ranges from 0 for no overlap to 1 for perfect fit. $R^2$ varies between 0 and 1 which a computed value of 1 indicates perfect agreement. A smaller RMSE indicates better agreement. These have been described in detail in the latest version of the manuscript (Section 3.3, Page 12-13, lines 263-276).

**Revisions in manuscript (RC1-Q4):** We added descriptions of the critical criteria used to assess HCA-FM model's performance compared to field observations:

(Page 9, lines 203-204) In the comparison of HCA-FM simulations with field observations, the visual image comparisons were made for the line or polygon features of inundation extent.

(Page 9, lines 206-208) The inundation extents simulated by the HCA-FM and survey extents are presented in Fig. 4

and Fig. 5. It is important to note that due to the limitations of rough survey data, validation was based solely on inundation extent.

**(5) RC1-Q5: Can authors provide insights into validating the HCA-FM model against regional field observations, including the specific data sources and methodologies used for comparison?**

**Response:** In the experiments of comparing HCA-FM simulations with field observations, visual image comparisons were made for the line or polygon features of inundation extent. It is important to note that due to the limitations of rough survey data, validation was based solely on inundation extent. Detailed comparisons for water depth had been made between HCA-FM and ADCIRC+SWAN coupled model. These have been described in the lastest version of the manuscript (Section 3.2).

The first experiment compared the inundation extent caused by Typhoon Lekima in Cangzhou, Hebei. Investigation team from the National Marine Environmental Forecasting Center investigated disasters around the south coast of the Bohai Bay. The second experiment compared the inundation extent caused by Typhoon Hato in Shenzhen, Guangdong. The Marine Monitoring and Forecasting Center of Shenzhen organized teams to investigate disasters in key regions. The field survey data in Shenzhen after Typhoon Hato included several locations that were severely affected, which indirectly reflected the inundation extent. These have been described in the latest version of the manuscript (Section 3.1, Page 7, lines 168-178).

**(6) RC1-Q6: What were the main findings regarding the ability of the HCA-FM model to reproduce observed inundation distributions, and how did its predictions compare to those of the hydrodynamic numerical model regarding inundation extent and submerged depth?**

**Response:** In Section 3.2, the simulated ability of HCA-FM is validated comparing its simulation with the observed inundation distribution, as given in Fig. 4 & 5, it is able to reproduce the actual inundation area for both two TC events. These have been described in the lastest version of the manuscript (Section 3.2).

In Section 3.3, as a result in the comparisons between HCA-FM and ADCIRC+SWAN (Fig. 6), the fit ratio δ of inundation extent was 0.92 for TC Lekima and 0.95 for TC Polly. For the submerged depth, the $R^2$ were 0.96 for both Lekima and Polly, the RMSE were 0.13 m for Lekima and 0.12 m for Polly. The results indicates a good consistency between the two models. In addition, sensitivity experiments associated with wind force and bottom friction were designed (Table 2), and the results shows that consideration of external forces ensures model's accuracy towards the storm surge inundation simulation. These have been described in the latest version of the manuscript (Section 3.3).

**(7) RC1-Q7: The study mentions significantly improved computational efficiency with the HCA-FM model, allowing for inundation predictions within a few minutes. Can authors discuss the implications of this improved efficiency for real-time storm surge forecasting and disaster management?**

**Response:** Hydrodynamic models are generally considered as unviable for areas larger than 1000 km$^2$ when the resolution required is less than 10 m. Forecast timeliness is often not met for larger scale forecasts. But the HCA-FM model requires significantly less computer effort than hydrodynamic models. Runtime savings portend that the model is suitable for large floodplains larger than 2000 km$^2$. In real-time storm surge forecasting and disaster management, the HCA-FM model can be used in conjunction with the hydrodynamic models. The HCA-FM can quickly forecast

the potential regions and hazard level affected by the storm surge to identify the most serious regions, which leaves more and sufficient time for the government to make decisions.

**Revisions in manuscript (RC1-Q7):** Supplements about the potential applications of HCA-FM were made in the revised manuscript:

(Page 17, lines 353-356) The reduction in computational time as well as its high stability and universality would have great significance in the practical applications of real-time early warning by rapidly identifying key affected areas on a large scale. HCA-FM can also leverage its computational efficiency to significantly reduce the time spent on probabilistic risk assessment that requires a large number of simulations, and contribute to more effective disaster prevention and mitigation strategies in coastal regions.

**(8) RC1-Q8: How does the high stability of the HCA-FM model contribute to its reliability in predicting storm surge inundation, particularly under varying environmental conditions and input parameters?**

**Response:** Since the iterative solving process of the HCA-FM is concise, there do not exist many constraints on the stability as those of the numerical model, which depends on factors such as the environment, mesh, and model parameters when solving complex differential equations. Therefore, different environmental conditions, grids and parameters do not affect the stability of the model, thus ensuring the reliability of the model in predicting storm surge inundation. As seen, we perform experiments conducted in different region, and the HCA-FM model performed well in all regions, including Laizhou Bay where it was tested against two typhoons with different tracks. These results suggest that the model is accurate and stable across different study areas and typhoon processes.

**(9) RC1-Q9: Considering the significant advancements demonstrated by the HCA-FM model, what are the potential applications at large space scales, and how might they contribute to more effective disaster prevention and mitigation strategies in coastal regions?**

**Response:** HCA-FM can be used not only for real-time disaster prevention and mitigation by rapidly identifying key affected areas on a large scale, but also for regional storm surge hazard and risk assessment. To conduct a probabilistic risk assessment, it is necessary to analyze the probability statistical characteristics of storm surge disaster-causing factors from their long-term time series. As long-term storm surge flood observations are sparse, a large number of simulations are required. Additionally, risk assessments often cover larger study areas, which can make running simulations using hydrodynamic models prohibitively time-consuming. Therefore, HCA-FM can leverage its computational efficiency to significantly reduce the time spent on probabilistic risk assessment and contribute to more effective disaster prevention and mitigation strategies in coastal regions.

**Revisions in manuscript (RC1-Q9):** Supplements about the potential applications of HCA-FM were made in the revised manuscript:

(Page 17, lines 353-356) The reduction in computational time as well as its high stability and universality would have great significance in the practical applications of real-time early warning by rapidly identifying key affected areas on a large scale. HCA-FM can also leverage its computational efficiency to significantly reduce the time spent on probabilistic risk assessment that requires a large number of simulations, and contribute to more effective disaster prevention and mitigation strategies in coastal regions.

**(10) RC1-Q10: Lastly, based on the findings of this study, what are the critical areas for future research or refinement of the HCA-FM model, and how might it be further optimized for broader practical use in storm surge prediction and coastal zone management?**

**Response:** Since computing the new state of a cell in CA depends only on the state of the neighboring cells at the previous time step, CA algorithms are well suited to parallel computation. It will be considered to apply parallel computation for HCA-FM model for further enhancement of computational efficiency in the future. Additional, consideration will also be given to designing an interactive operating system for the model to cater for more intuitive and easy use. In addition to improvement of efficiency, the principles of the model will also be improved in the future to take into account more comprehensive hydrodynamic mechanisms to improve the simulation accuracy. Relevant complement will be made in revised manuscript.

**Revisions in manuscript (RC1-Q10):** Supplements about the future outlook of HCA-FM were made in the revised manuscript:

(Page 17, lines 357-361) As CA algorithms are well suited to parallel computation, it will be considered to apply parallel computation for HCA-FM model for further enhancement of computational efficiency in the future. Additional, consideration will also be given to designing an interactive operating system for the model to cater for more intuitive and easy use. In addition to improvement of efficiency, the principles of the model will also be improved in the future to take into account more comprehensive hydrodynamic mechanisms to improve the simulation accuracy.

**RC2 from Anonymous Referee #1**

**(1) RC2: The authors have well addressed my comments. This manuscript is acceptable for publication in its present form.**

**Response:** Thank you for your comments and for recommending publication of the manuscript.

**3. Author's response and changes in manuscript to Comments from Anonymous Referee #2**

**RC3 from Anonymous Referee #2**

**(1) RC3-Q1: The developed model, called HCA-FM, uses a different approach for the wet-drying algorithm compared to the common CA-flood models, which typically employ the water balance approach. The novelty of this manuscript lies in the fact that HCA-FM utilizes the shallow water momentum equation to govern the transition rules for the CA model and incorporates external forces (e.g., wind stress and bottom friction). Therefore, the main focus of this study is to prove that by adding these new rules or equations, it will improve the accuracy of HCA-FM compared to common CA-flood models and its efficiency compared to conventional hydrodynamic models. However, based on this manuscript, the authors seem to fail to demonstrate this. The authors directly applied the model to a real event and compared the results with the ADCIRC+SWAN model.**

**Response:** Thank you for highlighting the importance of this study in the dynamic consideration and high efficiency of the newly developed model. The efficiency have been demonstrated in the manuscript by comparison with conventional hydrodynamic models (ADCIRC+SWAN), as detailed in Section 4 (Page 16-17, lines 341-352) ,and its dynamic consideration is evaluated by sensitivity study in Section 4 (Page 15, lines 309-322).

In order to more clearly demonstrate the significance of this work, we made some supplement description in the manuscript.

**Revisions in manuscript (RC3-Q1):** We made supplements to explain why we did not compare HCA-FM with common urban flooding CA models to illustrate the benefit of hydrodynamic consideration of both wind force and bottom friction on the accuracy of the simulation in the discussions:

(Page 15, lines 301-306) The innovation of HCA-FM compared to conventional CA models is its hydrodynamic consideration of both wind force and bottom friction. Storm surge inundation differs from urban flooding in terms of the geographical and meteorological environment. The CA models for urban flooding mainly simulate the spreading of fluids to lower elevation areas under gravity, whereas storm surge inundation usually occurs in coastal areas, where seawater spreads not only to lower elevation areas under gravity, but also to higher elevation areas under dynamic forcing. In view of this, it is not appropriate to use the common CA models to simulate storm surge inundation.

(Page 15, lines 307-308) Therefore, in order to illustrate the benefit of such considerations on the accuracy of the simulation, sensitivity experiments related to wind force and bottom friction were designed for two storm surge events.

**(2)  RC3-Q2: In my opinion, the authors could have used simpler examples (such as laboratory experiments or artificial cases) to prove the hypothesis. By using simpler cases, we can observe in detail what is happening within the model's results. From what I have seen, since the authors directly applied the model, we can only observe the comparison of flood extent between the model's results and observational data; therefore, the discussion is not deep enough.**

**Response:** This is a very good advice. We have carried out the following experiment to illustrate the effects of wind stress and bottom friction on inundation in detail:

Experimental setup: As shown in Fig. 7, a simplified typical terrain for storm surge inundation is designed with a constant slope (1:500), with the left side on the shore side and the right side on the sea side. The inundation source is located at the position where the ground elevation is 0, with a constant water level of 2.5 m and a constant onshore flow velocity of 1 m/s. For the onshore winds and bottom friction coefficient, we set up three sets of constant wind speeds of 0, 20, and 40 m/s (**where the wind speed of 0 m/s corresponds to the common CA-flood models**), and three sets of Manning's coefficient of 0.04, 0.06, and 0.08. According to the combination of the three sets of wind speeds and Manning's coefficient, the inundation simulations were carried out using the HCA-FM model.

[Figure]

Figure 7: Maximum water level distribution under different wind speed and Manning's coefficient conditions.

The experimental results are as follows: Figure 7 shows the maximum water level distribution under different conditions of wind speed and Manning's coefficient. The bottom friction prevents the seawater from propagating to the shore, and under the same wind speed condition, the larger the Manning's coefficient (the larger the bottom friction coefficient), the smaller the inundation area and the smaller the inundation depth; the onshore wind forces the seawater to propagate to the shore, and under the same bottom friction condition, the larger the wind speed, the larger the inundation area and the larger the inundation depth. It can be seen that the wind plays an important role in the storm surge inundation process, and if the effect of onshore wind is not considered in the inundation simulation, the potential inundation area and water depth will be underestimated. Thus, it is shown that it is necessary to consider wind stress and bottom friction in storm surge inundation modelling, which is the advantage of the HCA-FM model over the common CA flooding models.

Supplements are made in the revised manuscript.

**Revisions in manuscript (RC3-Q2):** The experiment was added in the discussion to illustrate the effects of wind stress and bottom friction on inundation in detail:

(Page 15-16, lines 323-329) In addition, to illustrate the effects of wind stress and bottom friction on inundation in detail, a simplified typical terrain for storm surge inundation was designed with a constant slope (1:500), with the left side on the shore side and the right side on the sea side (Fig. 7). The inundation source is located at the position where the ground elevation is 0, with a constant water level of 2.5 m and a constant onshore flow velocity of 1 m/s. For the onshore winds and bottom friction coefficient, we set up three sets of constant wind speeds of 0, 20, and 40 m/s (where the wind speed of 0 m/s corresponds to the common CA-flood models), and three sets of Manning's coefficient of 0.04, 0.06, and 0.08. According to the combination of the three sets of wind speeds and Manning's coefficient, the inundation simulations were carried out using the HCA-FM model.

(Page 16, lines 330-331)

[Figure]

Figure 7: Maximum water level distribution under different wind speed and Manning's coefficient conditions.

(Page 16, lines 332-340) Figure 7 shows the maximum water level distribution under different conditions of wind speed and Manning's coefficient. The bottom friction prevents the seawater from propagating to the shore, and under the same wind speed condition, the larger the Manning's coefficient (the larger the bottom friction coefficient), the smaller the inundation area and the smaller the inundation depth; the onshore wind forces the seawater to propagate to the shore, and under the same bottom friction condition, the larger the wind speed, the larger the inundation area and the larger the inundation depth. It can be seen that the wind plays an important role in the storm surge inundation process, and if the effect of onshore wind is not considered in the inundation simulation, the potential inundation area and water depth will be underestimated. Thus, it is shown that it is necessary to consider wind stress and bottom friction in storm surge inundation modelling, which is the advantage of the HCA-FM model over the common CA flooding models.

Thank you very much for your time and attention.

Best regards,

Xuanxuan Gao